# Location Intelligence Systems and Data Integration for Airport Capacities Planning

**Mirza Ponjavic [1] and Almir Karabegovic [2,*]**

[1]   Faculty of Engineering and Natural Sciences, International Burch University Sarajevo,
      71210 Ilidža; Bosnia and Herzegovina; mirza.ponjavic@gis.ba
[2]   Faculty of Electrical Engineering, University of Sarajevo, 71000 Sarajevo, Bosnia and Herzegovina
*    Correspondence: akarabegovic@etf.unsa.ba; Tel.: +387-35-363-110

**Abstract:** This paper describes an approach introducing location intelligence using open-source software components as the solution for planning and construction of the airport infrastructure. As a case study, the spatial information system of the International Airport in Sarajevo is selected. Due to the frequent construction work on new terminals and the increase of existing airport capacities, as one of the measures for more efficient management of airport infrastructures, the development team has suggested to airport management to introduce location intelligence, meaning to upgrade the existing information system with a functional WebGIS solution. This solution is based on OpenGeo architecture that includes a set of spatial data management technologies used to create an online internet map and build a location intelligence infrastructure.

**Keywords:** location intelligence; capacity planning; open source software; geographic information systems; data visualization; system integration; enterprise architecture; building information modeling

## 1. Introduction

Sarajevo International Airport (IATA: SJJ, ICAO: LQSA) is the largest airport in Bosnia and Herzegovina. It is organized in 10 sectors, 23 services, and six departments covering traffic and services in air traffic, commerce, engineering and maintenance, finance, legal, human resource and general affairs, information and communication technologies, security and protection, systems management, airport development, and cargo services.

Six separate companies have control over Sarajevo International Airport. For example, the refueling of the airplanea is entrusted to Energopetrol, an oil distributer company, flight control belongs to the Ministry of Transport, lighting of the airport runway covers Elektroprivreda, a power distributer company, while passport and customs controls are under the jurisdiction of the State Border Police. Further, as far as fire protection is concerned, the Fire Brigade positioned at the airport is under the control of the Sarajevo Canton Government, i.e. Civil Protection Service. The rest of the sector, like warehouse, cargo and other services within the airport building, are under the control of a firm called "Sarajevo International Airport."

The IT and Communications Technology Sector maintains the complete information system, consisting of several subsystems, such as the Departure Control System (DCS), Flight Information Display System (FIDS), and Fire Protection System (FPS).

Of the many systems that exist at Sarajevo International Airport, the DCS and FIDS systems have been specifically addressed. The Departure Control System (DCS) essentially serves the automatic processing of all key flight data. This process includes all necessary elements related to flights at the Sarajevo International Airport, such as departure, check-in, read-outs, load plans, capacity management, load balancing, airplane balancing, zero fuel bill calculation, etc.

FIDS is a display information system whose main function is to inform passengers about real-time flights. It is not limited to TV screens installed at airport terminals but can work also on websites and various mobile applications for Android, iOS, and Windows devices. The process of collecting information and broadcasting is presented in Figure 1. However, all the information on the TV screen is better seen in Figure 2, which shows the data model.

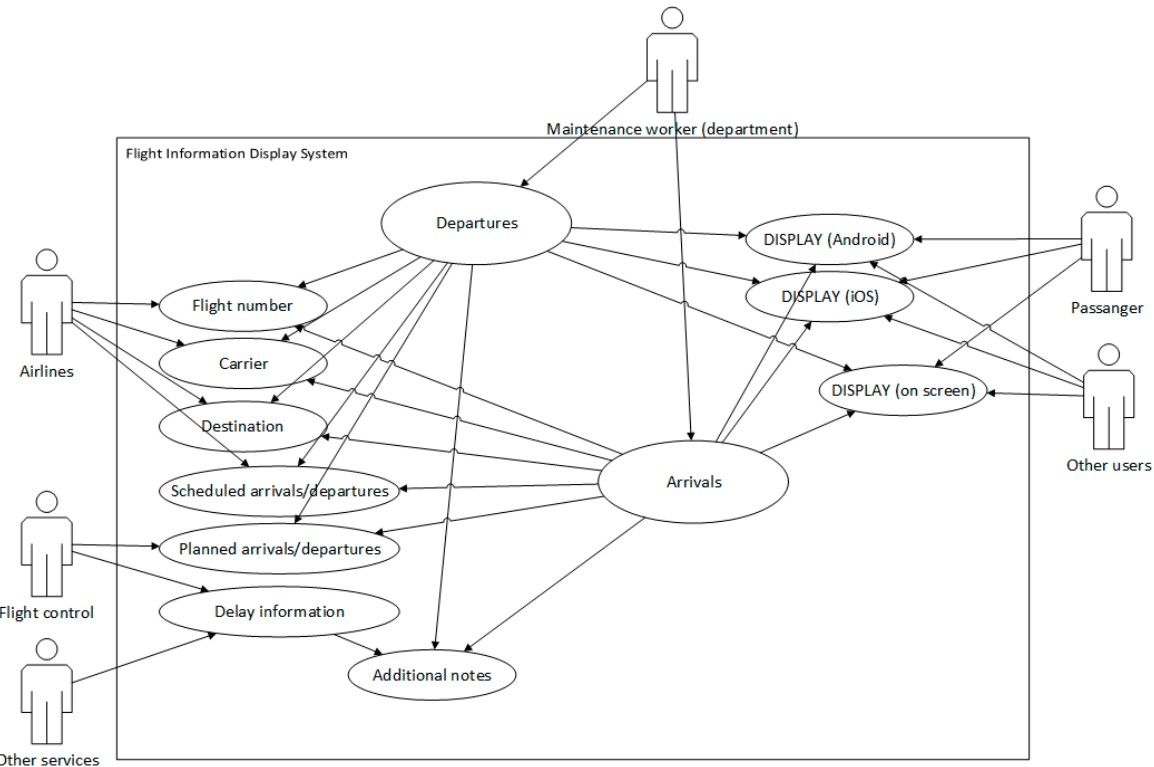

**Figure 1.** Use case of actual Flight Information Display System (FIDS).

The European Organization for the Safety of Air Navigation (EUROCONTROL, of which Bosnia and Herzegovina is a member) helps state regulators to develop a Terrain and Obstacle Data (TOD) policy according to ICAO Annex 15 on Aeronautical Information Services (AIS), which lays down the requirements for the publication of electronic TOD [1]. For the needs of the International Airport Sarajevo, with the aim to increase the degree of security of the various air operations, the Airport Obstacle Map—Type A was made. Airport has complete support for ICAO Annex 4 chart specifications, including the obstacle map (Figure 3), which verify data compliance for aviation standards like ICAO. The purpose of the obstacle map was to ensure conditions for safe movement of aircraft as well as to prevent disturbance of safety due to obstacles near the airport. Also, a database of obstacles that are in the plane of takeoff was made and geodetic map of the complete area of the airport within the fence was performed. Further, a 3D map of the airport (a database that should serve as the basis of the future geoinformation system) was created. Among other things, the location intelligence solution should provide management and maintenance of information related to such sets of maps.

Today's trends related to the wide availability of geoinformation have a direct impact on the quality of work of all the organizations and there is the growing need for: access to available geoinformation by all employees, centralization of data, improved interaction of employees (in terms of the mutual exchange of information, messages and tasks), and the introduction of unique methodologies and geoinformation technologies in all organizations. In order to use of available geoinformation and meet the operational needs of individual organizations, it is essential to support their business processes through the establishment of a centralized spatial information system.

Due to the frequent construction work on building new terminals to increase the existing capacities of the International Airport in Sarajevo, as one of the measures for more efficient airport infrastructure management, the development team suggested the introduction location intelligence, i.e., upgrade of the existing information system with the functional WebGIS application [2].

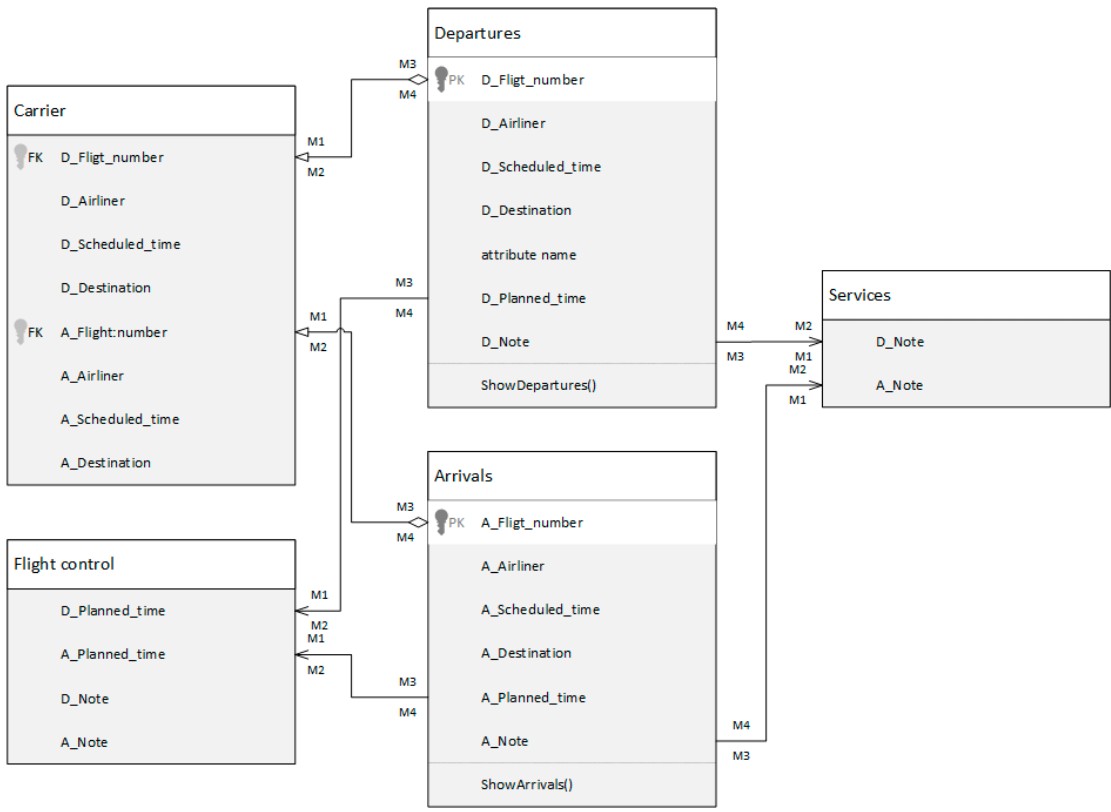

**Figure 2.** Data model of actual Flight Information Display System.

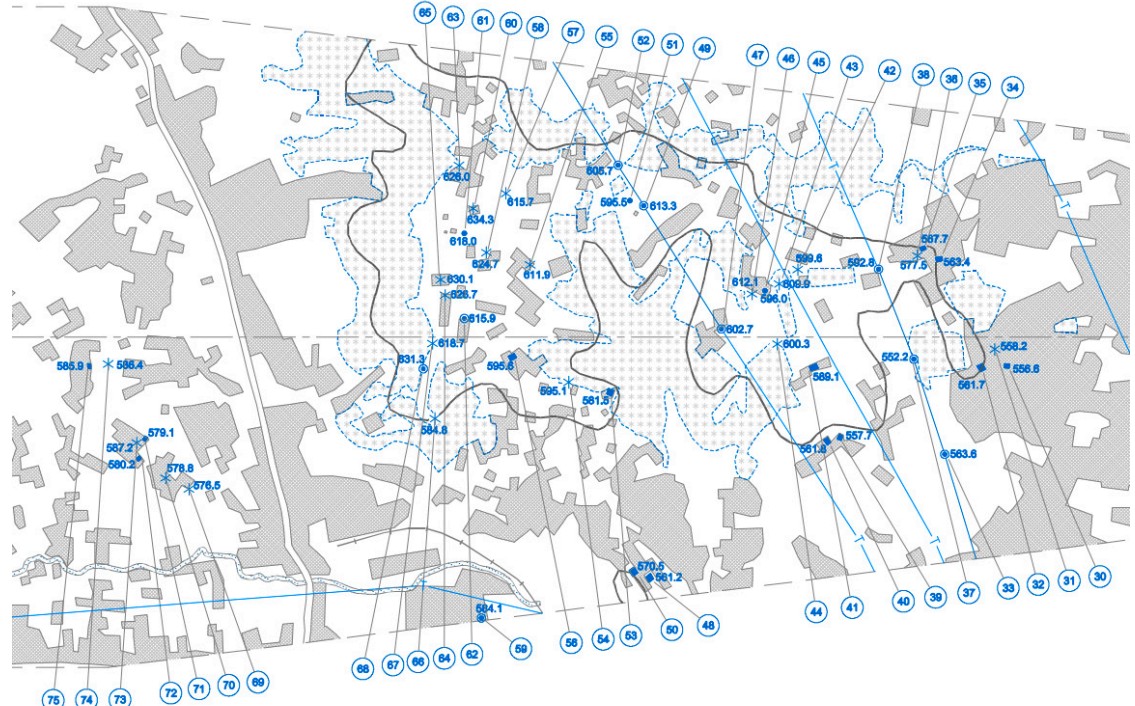

**Figure 3.** Airport obstacle map—ICAO type A (operating limitations).

The use of location intelligence enables the strengthening of technical and human capacities in terms of more efficient implementation of regular activities, thereby contributing to a better internal organization, providing quality services to all users and creating conditions for further development. In this case it is particularly emphasized the need for more effective planning and monitoring of the further development and the actual construction of airport infrastructure due to increasing transport capacities.

Also, due to the intensive construction work and the movement of construction equipment at the aerodrome, development team has recognized the need to enhance the control of the staff movement and access to aircrafts parking and runway area, as well as their readiness to react in emergencies. Various research programs help airports identify needs and assess current capabilities with respect to using geographical information systems (GIS) in emergency management (EM), emphasizing that "Airports primarily use GIS for asset identification, or the understanding of where things are and how they can be accessed to make decisions faster in emergency situations" [3].

Airport enclosure space is composed of the following three areas:

- The area in front of the airport including parking for passengers, employees and taxi rankings;
- The area with airport buildings including passenger sector (check-in, passenger services, reception and take-over of luggage, reception area) and the control (administrative) sector with zones for employees and persons with special permits (border services, IT sector, flight control, metrology stations, fire brigade units, fuel refueling services, lighting service, warehouses, etc.); and
- The area for aircraft, with runways and parking.

The airport information system includes a subsystem with a TETRA solution for locating internal communication stations. Staff equipped with these cells have limited movement in the area in front of the airport and within the control sector. Information on the position of each aircraft on the parking lot is registered in the FIDS system. The space for moving and parking the aircraft is under the supervision of the security and protection sector, which includes monitoring the movement of wild animals. To make spatial decisions on emergency action, information on staff locations, aircraft position, wildlife movement, and other related information should be geovisualized. This implies the use of dynamic web services in an innovative manner to support mechanisms for accepting and simultaneous presentation of data selected from multiple information subsystems.

## 2. Introducing Location Intelligence

Based on a preliminary analysis provided by management team, it was concluded that it was necessary to upgrade the existing information system with components for spatial data visualization and monitoring changes in space related to further construction of the airport. These components imply a GIS (Geoinformation System) implementation that is the continuation of the implementation of planned IT projects within the organization [4–11]. GIS implementation generally means the following steps [8–11]:

- Choice of GIS platform (client, database server, application server);
- Inventory of available data in an organization;
- Identifying external data sources and access method (web services);
- Creating and implementing a unique spatial data model;
- Establish rules and procedures for working with spatial data;
- Definition of analytical functionality;
- Web site development for spatial data presentation; and
- User training.

For presenting spatial data in real-time using an online interactive map, as well as searching, analyzing and reporting, it is necessary to centralize all spatial data and provide permanent access to spatial data from other institutions and organizations. In this sense, "centralization and integration

of data and processes through a web-oriented spatial information system as an integral part of the airport information system" is imposed [4–8].

The establishment of the geoinformation system and the creation of a unique database will result in rationalization, integration, and decision-making optimization in multiple sectors, as well as more efficient data exchange within the entire enterprise. Therefore, in addition to the services of provision and installation geoinformation technologies, this project includes activities to establish and maintain the spatial database, the development and integration of the application solution, the preparation and migration of digital data, and various accompanying consultancy services.

In this sense, the development of this system can be seen through the following phases:

- Integration, centralization, and presentation of all spatial data;
- Development of an integral WebGIS application that will allow access to all available spatial data sets;
- Installation, tuning and testing of the system including user training;
- System maintenance and technical support.

The implementation of the system in the narrow sense includes the following activities [9–11]:

- Modeling and reengineering of business processes (related to spatial data);
- Functional and non-functional specification of the system;
- System design: system architecture, software architecture and data model;
- Installation of software licenses;
- Implementation (development);
- ETL of existing data;
- Metadata entry;
- User and administrator training; and
- Maintenance.

## 3. WebGIS Development

The architecture of spatial information system includes organizational concept of the technical components of the system such as networks, hardware, software and databases. A corporate solution is proposed for the International Airport Sarajevo, which implies:

- That spatial information is centrally located;
- Shared access to data (collected once, used multiple times);
- Reducing data redundancy;
- Always use up-to-date data;
- The possibility of extending the use of spatial information from other organizations; and
- Integration of spatial information system with other information systems in the organization.

It is understood that, within the administration of the data, it will be provided system security (access privileges) data standards and supporting documents. Also, during the implementation, it is necessary to consider the existing way of managing information in the organization.

### 3.1. Information Management in the Organization

Considering that GIS is a complex system for managing spatial and attribute information at the level of the whole organization and that International Airport Sarajevo as a public company is certified for ISO 27: 001–defining Information Security Management System (ISMS), it is necessary to define information management related to the project implementation. Information management at the enterprise level is defined by documents like statement of applicability, classification, marking, and use of information, and information asset management.

According to Information asset management document, process owners are responsible for maintaining a list of information assets, so it is imperative that the owners of the information give consent to use the information in the GIS. Given that they manage the information, they are the ones who will monitor the change of information and submit them to the GIS administrator for update. Overall information management also implies sharing of spatial information with other organizations in accordance with ISMS.

*3.2. System Architecture*

The WebGIS solution is based on OpenGeo architecture (Figure 4), which represents a set of technologies for creating online internet map and building IT infrastructure. The foundation of this architecture is the use of functional layers:

- Data layer;
- Application (logical) layer; and
- Presentation layer.

GeoServer is selected as a server that generates maps and controls spatial data. Geoserver implements OGC standards—WMS, WFS, WCS. The user interface allows easy adding, updating, deleting data sources and layers, and setting styles (Figure 5). GeoServer has a built-in GeoWebCache tile cache server that speeds up the display of maps.

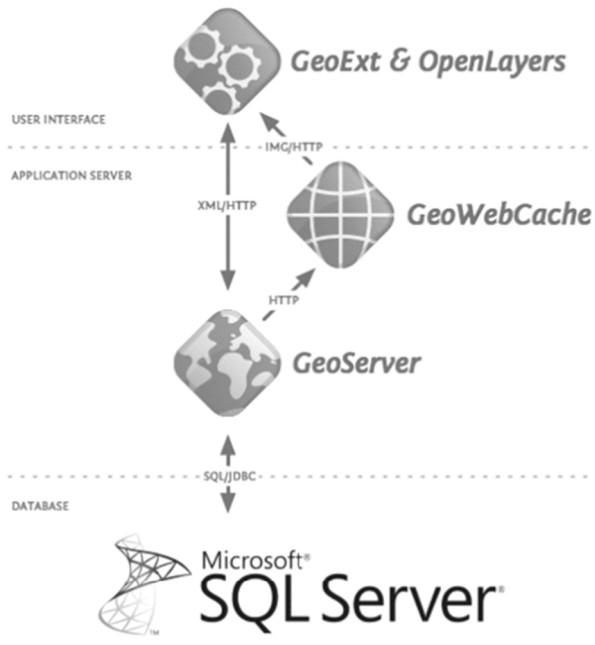

**Figure 4.** System architecture, modified from OpenGeo architecture, Source: OSGeo [2].

For the development of front-end application, it has been selected OpenLayers—the JavaScript library for browsing and interacting with spatial data via web browsers. OpenLayers enables easy integration with public web services. The second component is the GeoExt—JavaScript framework for building rich web applications that have desktop application performances while the data presentation is fully executed in the web browser.

For the data layer, the WebGIS solution uses the latest stable version of the DBMS Microsoft SQL Server. Full OpenGeo architecture uses PostgreSQL with PostGIS as data layer, but users insist on Microsoft technology as that which is most familiar and trained for.

As a desktop tool, MapInfo Pro (Figure 6) was chosen, which is a standard software for desktop cartography, visualization, and geographic analysis. Direct read/write capabilities in databases through ODBC connectivity allow users to keep all organization data up-to-date.

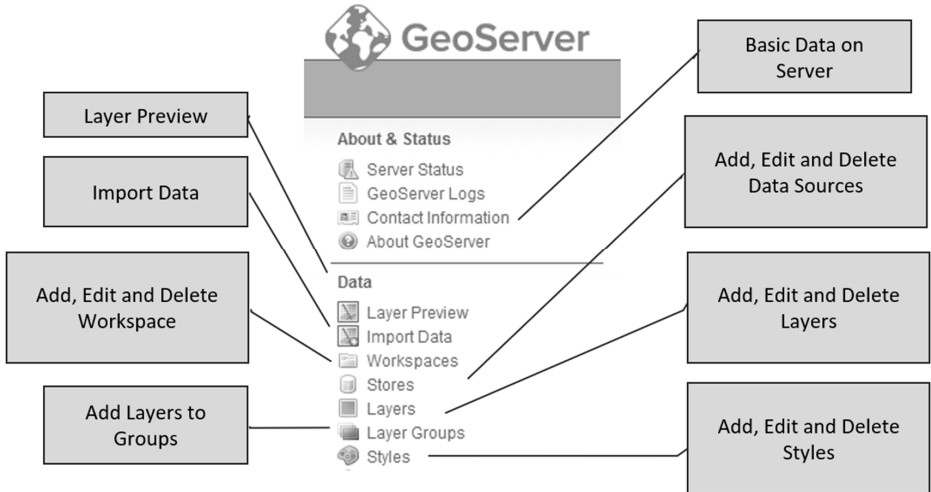

**Figure 5.** User interface (GeoServer).

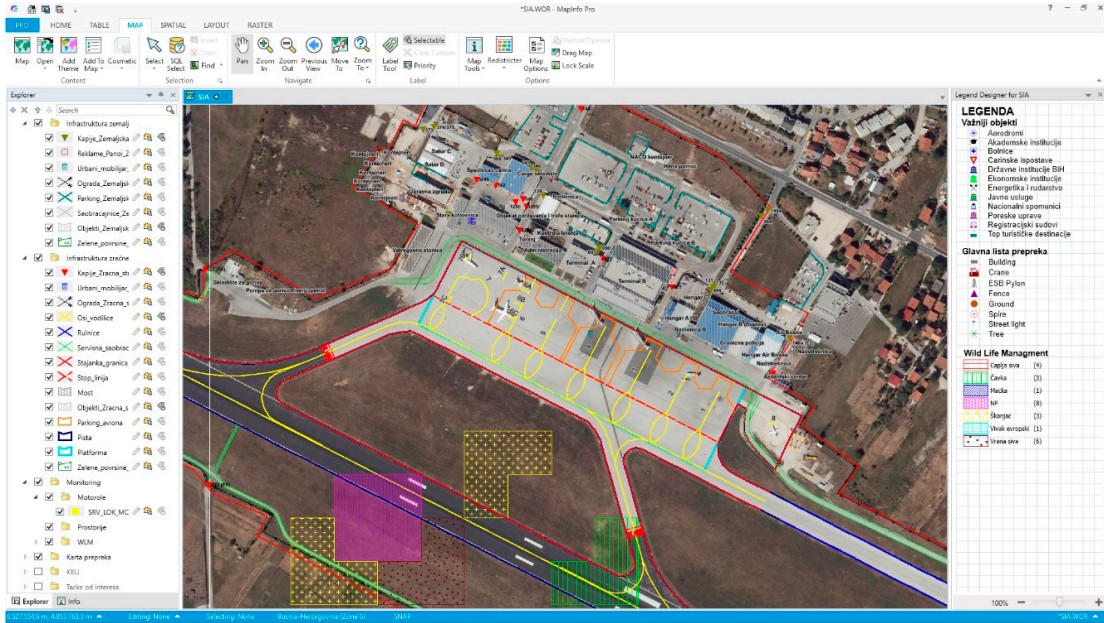

**Figure 6.** MapInfo Pro desktop tool for spatial data administration.

### 3.3. Data Modelling

The most extensive part of the work in this step was the development of data model, which ensured that all available geometric, topological and descriptive data (Figures 7–9) were placed within a logical model implemented in the DBMS user environment. In this way, spatial data is fully integrated with other data and is used through applications that are not specialized for GIS. The proposed graphical and alphanumeric data model is consistent with the cadaster database model and the metadata catalog based on Core Metadata for Geographic Datasets—ISO 19115: 2003 (E) standard.

The geometric data model is based on available geometric types of GML 3.1.1. standardized geometric data model (spatial topology), as well as associated complex types that enable the maintenance of GIS data using a time-domain (time topology) according to ISO 19107: 2003 (geographic information—spatial schema) and ISO 19108: 2002 (geographic information—temporal schema) standards. The implementation is performed through Feature Data Objects (FDO) as the basis for a standardized classification of graphical and non-graphical data, respecting the OpenGIS specifications that describe interfaces (for deployment) and formats (for exchange) of data collections as well as their adaptation with current technologies.

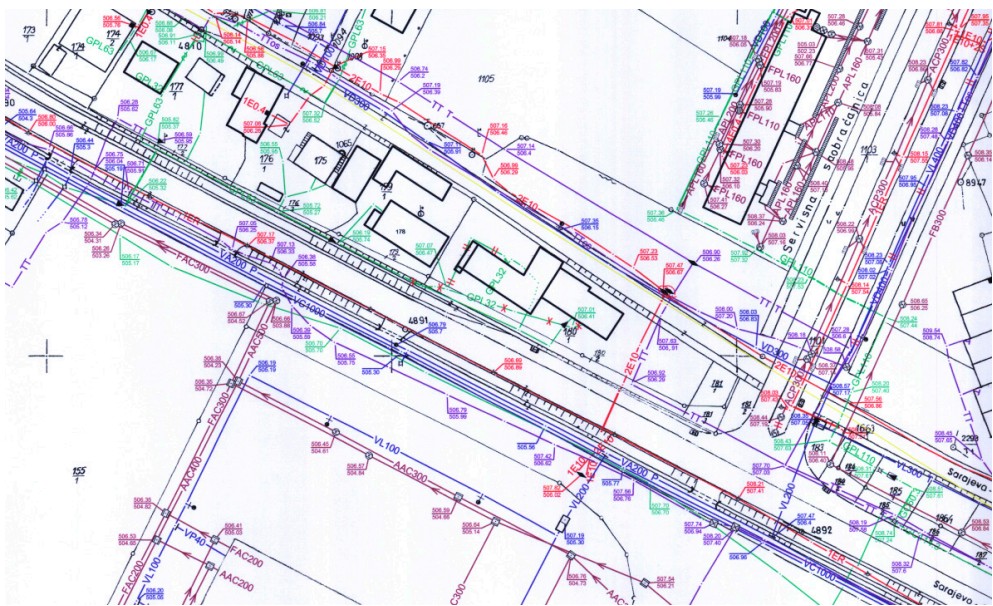

**Figure 7.** Available spatial data sets (cadaster of underground utilities).

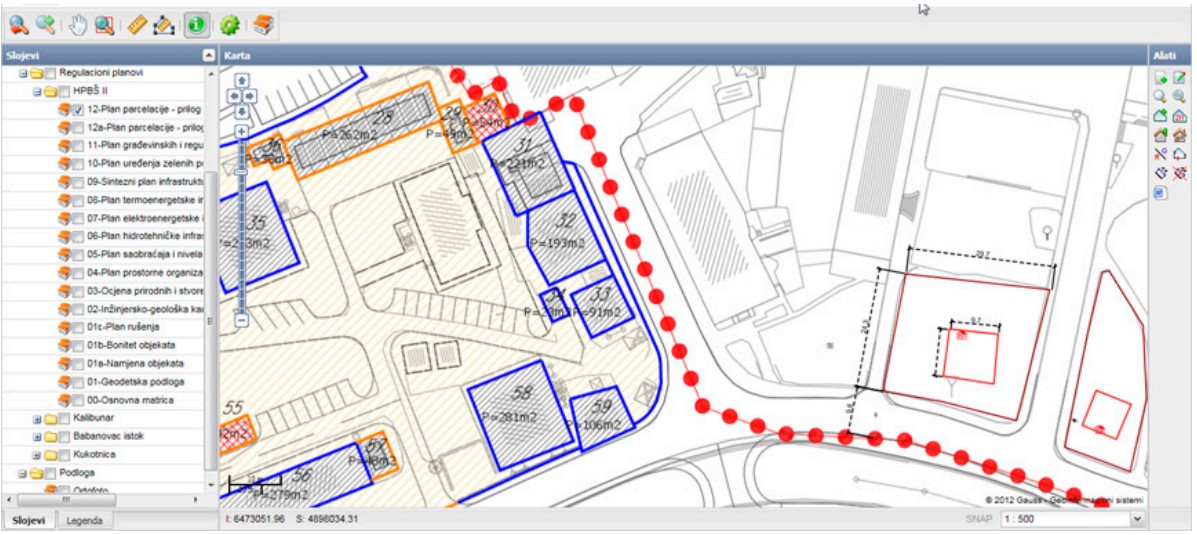

**Figure 8.** Available spatial data sets (regulation plans).

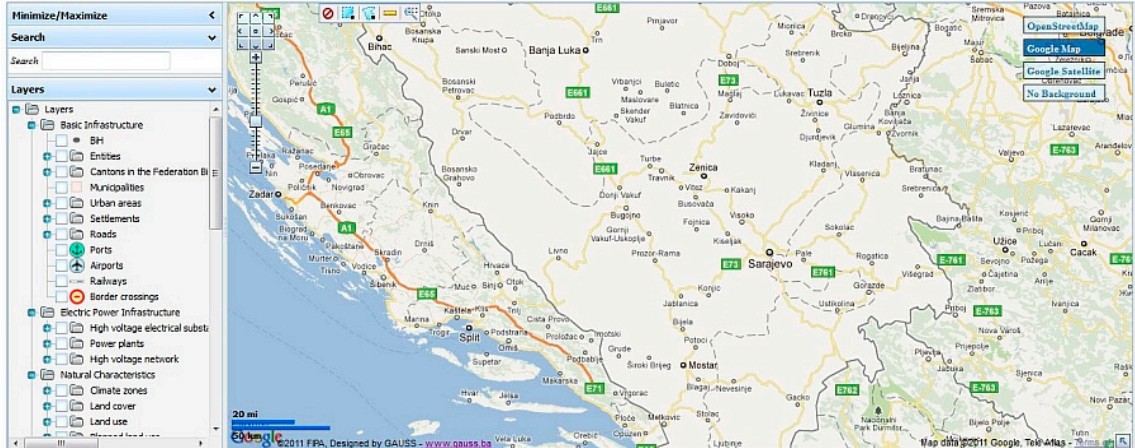

**Figure 9.** View of general geographic data.

### 3.4. Centralization and Integration of Spatial Data

Under the centralization of existing data sets, it is considered the collection, structuring, and placement of all existing spatial data on the central server, i.e., in the unique database to allow access and insight for all employees within the organization.

Integration and centralization of data were related to the following spatial data sets:

- Obstacle map—type A;
- Database of obstacles that are in the range of takeoff plane;
- Geodetic map of the complete airport area;
- Parking map;
- Available orthophotos;
- Regulation plans (Figure 8);
- Urban plan;
- Set of general geographic data (Figure 9); and
- Digital terrain model (DTM).

Within the project implementation, migration of all spatial data into a new accepted data model was performed. Among the data formats, dwg, xls, pdf, and doc formats were prevalent. It was necessary to validate some of the data in such a way as to perform their geodetic control on the ground, to survey the missing characteristic objects and to estimate their accuracy. The second part of the validation was carried out after the introduction of up-to-date geodetic database and comparing the data.

System integration with other subsystems is done based on web services. There is list of such services like DCS, SITA, WorldTracer, FIDS, and almost every office and official room of Airport building with list of equipment per room. An example is the web service for equipment per room:

**Web Service Code Template.** *http://domain:port/schema/table/\protect\T1\textbraceleftcode\protect\T1\textbraceright*

| where | domain is Domain Name Server, e.g., gisws.sia |
| --- | --- |
| | port is assigned port number, e.g., 89 |
| | schema is DBMS schema, e.g., Room |
| | table is tabke name from DBMS schema, e.g., EquipmentByRoom |
| | {code} means room code, an attribute from table, e.g., 03032006. |

### 3.5. Testing

Before use, all new system functions have been tested. The following tests were carried out: verification of functionality (presence, correctness and functional suitability, contradictions, etc.), ergonomic testing and run time (performances, reliability, ease of use, maintenance, documentation etc.), examination of formal properties (compatibility, interoperability, compliance with standards, compliance with internal rules and legal regulations, software quality, etc.), and security testing. The purpose of the user acceptance tests (UAT) is to confirm that the system is developed in accordance with the requirements and is ready for operational use. The testing phase is divided into three parts:

- Internal test by the development team;
- Beta test by the beneficiary;
- Acceptance test by the beneficiary.

Users were able to test the application and make their comments through the development phase since the staging server was configured at the beginning of the project and updated frequently with new features. A master test plan, which includes test cases, scenarios, and test logs, is developed providing improved communication among all parties involved in testing.

Here, different types of tests were done. Internally, the development team did inspection code, walkthroughs, and desk checking. Before deployment, unit and integration testing where modules

are integrated in a top-down incremental fashion were done. At the end, together with the airport development team, complete system testing and integration with other systems was performed.

Based on functional requirements, test cases had following elements:

- Test case number;
- Functional requirement description;
- Date of testing;
- Short description of testing;
- Scenario of using system;
- Step-by-step, result, status, and comments of testing;
- Actual vs. Expected output of testing application; and
- Signature of person doing testing.

Users tested pilot versions according to detailed technical specifications and test plans, and scenarios were successfully completed. After the summary report of the test phase was completed, the management of the airport gave the approval for the use of the system.

## 4. Discussion

The focus of this work is the application of location intelligence (tools, techniques, and methodologies) with data integration and system integration for airport capacities planning. By developing the geoinformation system for the needs of the International Airport Sarajevo, it has achieved the intention to introduce location intelligence as a new paradigm of using spatial and non-spatial data. In this way, it is also ensured better exchange and more efficient information management between individual aerodrome services. The main purpose of the WebGIS solution was the presentation of spatial data to users at the level of the entire organization, including the dissemination of vector and raster data of airport infrastructure, cartographic thematic data, and textual data with tables, diagrams and various documents. Additionally, information feedback from the system is continuously provided to users who make decisions based on location intelligence. Through the intuitive user interface of WebGIS solution it is enabled access to a wide-ranging system functionality including search and selection, distribution, publishing, administration and maintenance of spatial and non-spatial data relevant to airport capacity planning.

### 4.1. System Integration

Components at systems and application level for Enterprise Architectures [12–15] are beside operating systems and software applications, also Web Services and Service Bus. Today, web services use open standards protocols (like SOAP) and standard formats (like XML). It could be link to any information resource (file system, database, web application) and is predefined to work on the web. Service-Oriented Architecture is answer for building software applications that use services available in a network or the web. It is the trend that building loose links between software components, which also promote concept of reuse.

It is obvious that in an enterprise architecture such services could be huge amount and to make order SOA include term Service Bus or Virtual Enterprise Network, which is basically independent platform that catalogues all available service and allows any system to interoperate with any other system. Figure 10 shows whole picture integration of all web services for airport enterprise architecture using Universal Description, Discovery, and Integration (UDDI) method as metadata service and SOAP messages.

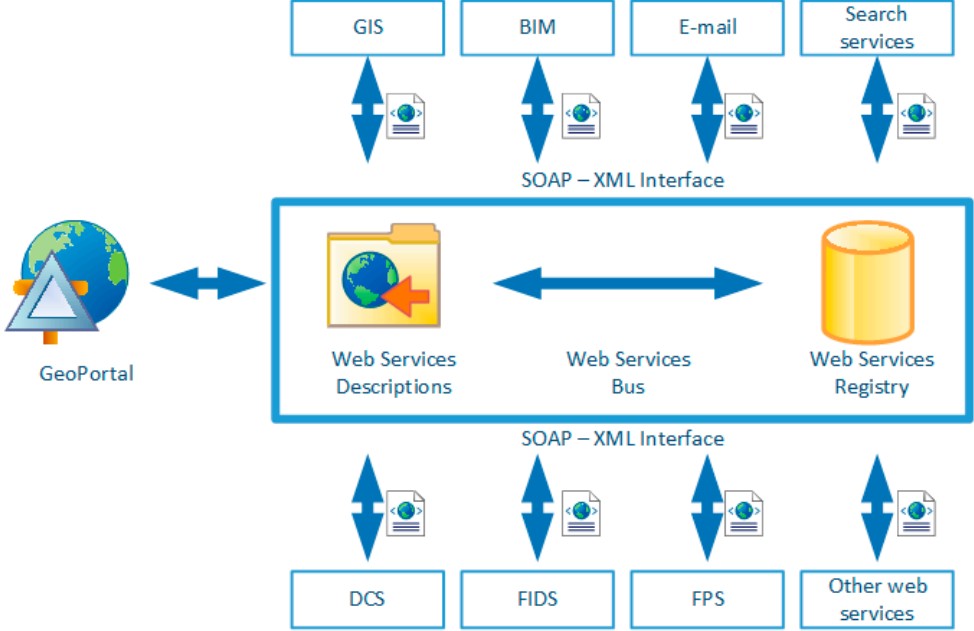

**Figure 10.** Integration of set of web services for airport enterprise architecture.

### 4.2. Location Intelligence Application

The geographic information system is the integral part of the complete location intelligence system. Its presentation component is a very important part of the developed user interface (Figure 11) because users have the need to review the data, set queries and operate with maps and other spatial and non-spatial information. It allows to display spatial information, map rendering, editing, queries, and spatial data analysis (Figure 12).

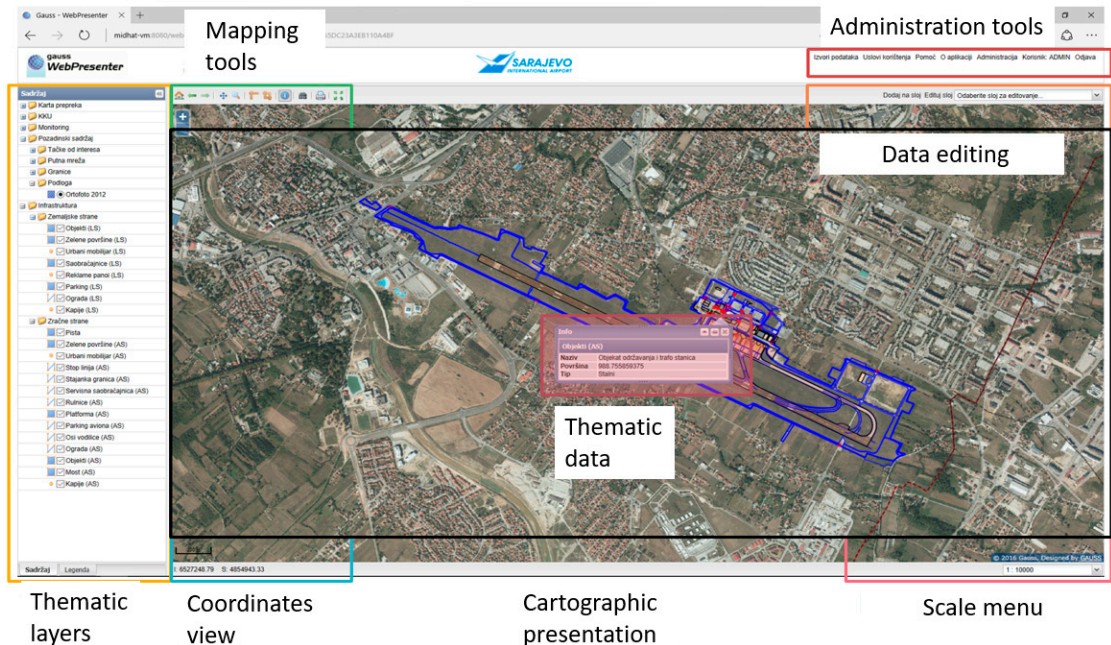

**Figure 11.** WebGIS application interface.

In this way, the realized location intelligence enables a complete and constant spatial overview of the airport infrastructure development throughout all phases of its planning, design, construction and use. Improved visibility of the airport property was provided. Assets information can be accessed

from each location and by any user (project manager, airport administration, contractor, supervisor, maintenance officer, security guard or airport service user) by smartphone or tablet.

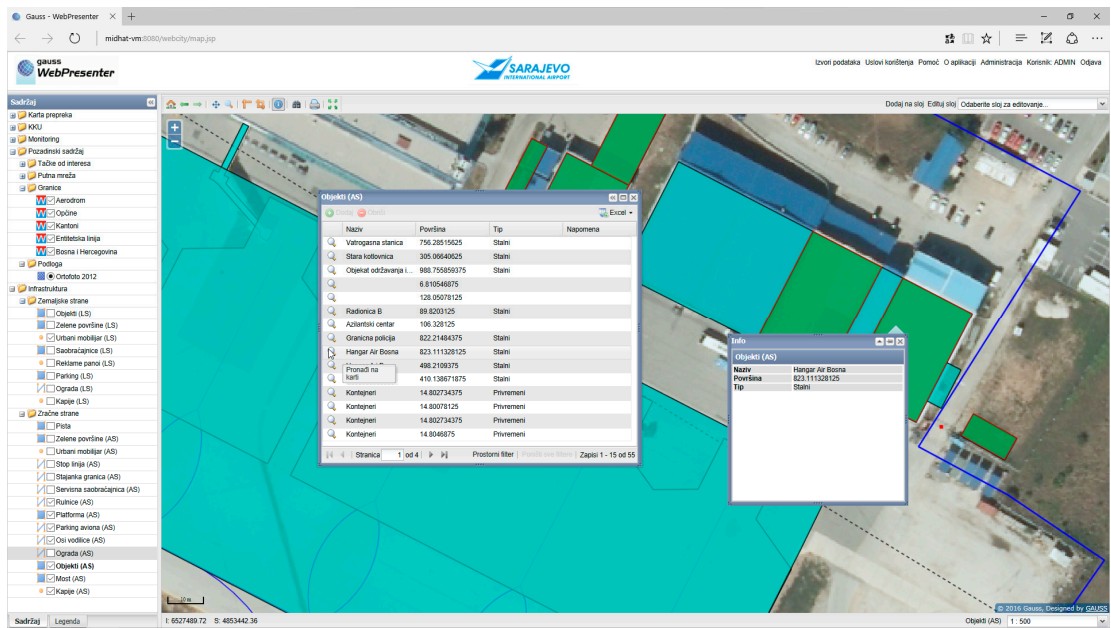

**Figure 12.** Spatial queries and search data functionality.

Location intelligence has increased the efficiency of asset management, enabling the display of every part of the asset, both for daily inspections and for periodic monitoring of infrastructure. Information on asset status is shared across the organization for easier decision making of maintenance teams. This increases the life span of equipment - from lifts to moving lanes in terminal buildings and equipment around them. When inspecting, with access to the infrastructure database by smartphones, it is possible provide data updates directly in the field, thereby reducing the errors.

With the location intelligence use and by knowing the exact location of underground utilities infrastructure, all expensive breaks could be avoided. Monitoring and registration of related spatial phenomena make it easier to recognize and understand the causes of breakdowns and damages, which, together with the preventive maintenance of the infrastructure, reduces unnecessary costs for its repairs.

A live integration consists of real-time requests from WebGIS to a remote system and updating response information before displaying them in the frontend. In the project, there were few such integration, but most innovative were visualization of data selected from TETRA, FIDS, and Wildlife Hazard Management System including:

- Airport staff location using location from Motorola two-way radios in real time, and geofence as predefined set of boundaries (boundaries of movement zone);
- Aircraft location on the parking positions, in aircraft service areas, taxilanes, etc. With different symbols for visual recognizing of the situation on apron;
- Wildlife hazard assessment for movement of wild animals (birds, mammals, and reptiles) and observation of different types of wildlife associated with safety issues at airports; and
- Other features related to decision making in emergency situations.

For all this dynamic information, there are set alarms for leaving these zones or abnormal happening. But most interesting is that for first time dispatchers and other management personnel can see in an intuitive way all together on one map which dynamically shows changes and alarms even without user interaction. In addition, it is possible to provide visualization of air quality, wildlife population monitoring, bird migration, noise level etc.

The airport information system expanded with the location intelligence component and dynamic web services providing spatial data visibility in real time and enables tracking of staff communication devices using geofence functionality so staff can move only where it is needed. This system can be further successfully applied to airport security, because it can serve as a platform for the integration of digital video records, sensors, real-time monitoring and management with security personnel, ensuring permanent visibility and all airport areas monitoring. It is possible to integrate other systems, such as security system, access control system, CCTV system, property tracking system, anti-fire system, etc. to prevent dangerous events and protect the safety of passengers, employees and property.

According to frequency of flights and the number of passengers, the airport in Sarajevo belongs to smaller airports. Purpose of this paper is to demonstrate that performance of developed LI solution with application of OSS can meet the needs of this type of airports. Benefits and limitations in terms of its use and maintenance are mainly related to general OSS application issues. The applicability of this approach to the development of similar solutions for larger airports that mainly use commercial software depends on many factors related to the number of employees, transport capacities, implementation time, developed system reliability, maintenance costs, hardware requirements, complexity of the solution for use and maintenance, etc. In general, it may be appropriate for lower maintenance costs, easier use and faster user acceptance, which is particularly significant if there are limitations in the number of employees in IT department of airport.

### 4.3. Data Integration

The establishment of a geoinformation system enabled wide availability of many spatial data sets including obstacle map, geodetic, ortho-photo, spatial planning, geographic data, digital terrain model, and many other digital data (Figure 13). However, several spatial data sets documenting airport infrastructure, such as project and contract documentation including building intersections, architectural solutions, installation views, mechanical drawings, construction data, technical specifications, etc. are still only available in analog format. It reduces the efficiency of their use in terms of access, manipulation, distribution and duplication. This data is of particular importance for the further development of infrastructure and services related to its maintenance and modernization, as well as for the construction of the new facilities. By establishment of digital archive system, it would be easy to handle this information through spatial search of the project documentation, as well as their further updating.

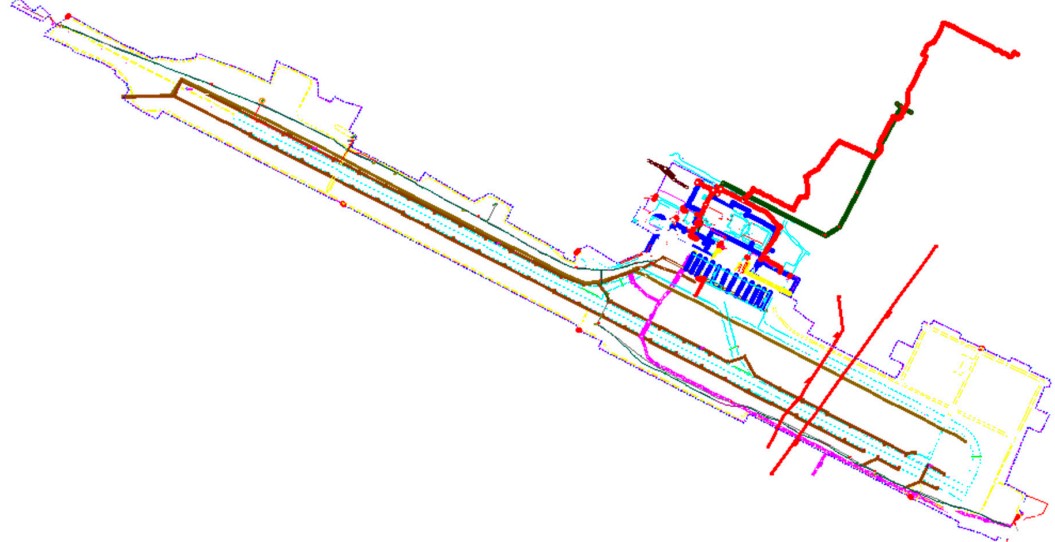

**Figure 13.** CAD presentation of Airport.

In that sense it is necessary to provide

- Inventory and digitization (scanning and spatial referencing) of all spatial data relevant to the maintenance, arrangement, development and upgrading of existing infrastructure, and construction of new ones;
- Integration and centralization of archival data in one common database;
- Development of web application for the presentation of all sets of archive spatial data that will allow access to all services that need this data;
- User training and maintenance of the system with technical support.

In addition, such digitized drawings can be used for development Building Information Modeling (BIM) system that should provide a three-dimensional presentation and models for maintenance purposes [16–20]. Development of the digital archive system and BIM, and their integration with GIS could be the next phases of information infrastructure development to support the construction and maintenance of airport facilities. Along with advanced 3D graphics and BIM, future design of sustainable capacity expansion projects will be facilitated, which was the main goal of location intelligence introduction for the airport.

BIM introduces several relevant engineering data for buildings. Through the establishment of a unique project data source, BIM technology seeks to achieve the integration of distributed heterogeneous engineering data and support the sharing of information and collaboration between different participants and different phases in the life cycle of the building.

With addition of BIM technology, the whole system is enriched with many tools, not standard for the GIS project, such as the amount of take-off, cost estimation, construction security analysis, workspace conflict and four-dimensional with different functions, such as building simulation, scheduling, resource management and site management. It also includes linear scheduling, Gantt chart, network diagram, tree structure, and 3D graphics platform for visualizing temporal, relational, hierarchical, and spatial information to support the construction of multiple systems.

In the operation and maintenance phase, it is necessary to use standards for data description, a data model with a uniform modeling language (UML), which help to more effectively manage life cycle information, including 3D visualization of buildings. The use of Level of Details (LOD) concept in project relates to model details, which describes the BIM component from the lowest conceptual design level to the most developed modeling level. There are five levels: LOD 1, LOD 2, LOD 3, LOD 4, and LOD 5 that refer to the conceptual design model, developed design model, presentation model, construction model and built model (Figure 14). Many researchers and practitioners have studied how to exchange information between BIM and GIS and how to solve all the differences, but it is still very difficult to share 3D information among different users or databases in different operations, from planning, design and construction to maintenance. There is no optimum and unique conversion due to different approaches to modeling these two information models.

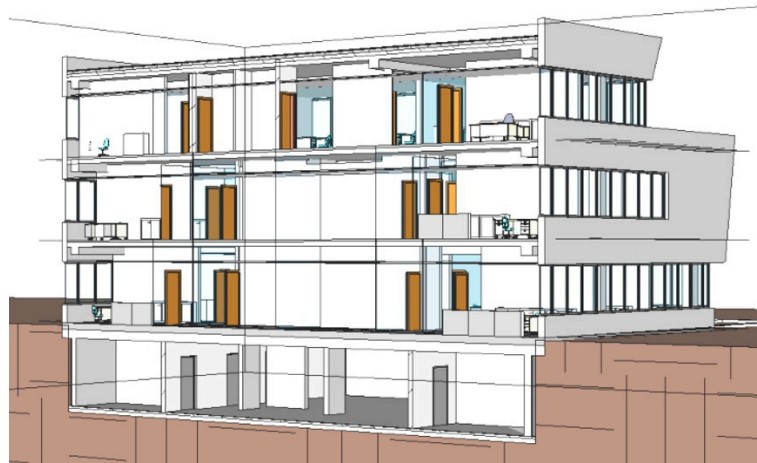

**Figure 14.** Building Information Modeling presentation of Airport building.

Converting a BIM model into a GIS model implies the simplification and removal of details and inadequate information in the data. A reverse operation, from GIS data to BIM is considered less useful and involve adding more details to the data. The biggest problem with data integration is that BIM models can contain many geometric and topological errors, which is very problematic in GIS applications, since spatial analysis involves complex operations such as Boolean setup operations.

## 5. Conclusions

This paper presents the approach for establishing a location intelligence solution for the needs of the International Airport in Sarajevo based on OpenGeo architecture. Through the analysis of the existing information system it has recognized the need for its extension by the WebGIS component. Its implementation is described through activities: integration and centralization of data, development, testing and commissioning of the entire system. Along with the architecture solution, a description of applied technology, data modeling and web services is given. The solution has the functionality customized for collaborative environment (cooperation between several internal/external sectors and organizations).

- Multiple access to web services;
- Full interoperability (simple updating of data from most GIS/CAD or Office tools);
- Multilingual interface.

    The presented WebGIS solution

- Offers the organization a complete solution for presenting and sharing business spatial data by the Internet/Intranet;
- Allows to monitor spatial changes in close real-time;
- Helps to design and realize ideas;
- Provides location-conscious decision-making and business information with spatial dimensionality; and
- Provides spatial visualization of ideas, plans and business potentials,

Thus, it provides more efficient management of airport infrastructure for the maintenance and expansion of its capacity.

One of the requirements of the development team is to provide IT support to enhance the control of the staff movement and access to aircraft parking and runway areas. For the GIS presentation of staff location and other features related to decision-making in emergency situations, the dynamic web services are applied in an innovative manner. This implied the development of special mechanisms for accepting and simultaneous presentation of data selected from multiple information subsystems.

The integration of BIM and GIS data is a complex topic [21] that is dealt with in different ways. This includes semantics and geometry and can also involve the conversion of BIM and GIS data either to a unique model or to standards of each other. Basically, the problem is in the roots of both models, where the roots of BIM are CAD graphics that support more geometry types and for more visualization purposes does not require special topological correctness, while GIS is based on DBMS asking to avoid redundancy and inconsistency, i.e., geometrical and topological errors to be able to run spatial SQL statements.

One of conclusions by the authors of this research is that much of this data is already passed from one model to other in some moment, as some data in the GIS model have their roots in BIM/CAD like parcels, and the reverse, such as infrastructure objects. This calls for deeper integration from beginning and working together in both models. This trend is exceptionally visible in fact that two ISO committees have established close cooperation: ISO/TC 211 Geographic Information/Geomatics Committee and ISO/TC 59/SC 13 Organization of Information on Construction Works; to create ISO/AWI 19166 Geographic information—BIM to GIS conceptual mapping (B2GM).

As a method to achieve the integration, the usage of the Web Service Bus as part of Enterprise Architecture is here suggested.

**Author Contributions:** Conceptualization, M.P. and A.K.; Methodology, M.P. and A.K.; Software, A.K.; Validation, M.P. and A.K.; Formal analysis, M.P.; Writing—original draft preparation, M.P. and A.K.; Writing—review and editing, M.P. and A.K.

**Funding:** The APC was funded by Gauss d.o.o. Tuzla, Bosnia and Herzegovina.

**Acknowledgments:** We would like to thank our colleagues from GAUSS d.o.o. Tuzla and the Geospatial Research Center in Sarajevo for technical support and transfer of materials and project documentation we used to write this paper. We would also like to show our gratitude to the project development team of Sarajevo International Airport for the extraordinary cooperation during the realization of the project used for case study of this research.

**Conflicts of Interest:** The authors declare no conflict of interest.

## Abbreviations

The following abbreviations are used in this manuscript:

| | |
|---|---|
| AIS | Aeronautical Information Services |
| B2GM | BIM to GIS conceptual Mapping |
| BIM | Building Information Modeling |
| CCTV | Closed Circuit TV |
| DBMS | Database Management System |
| DCS | Departure Control System |
| DTM | Digital Terrain Model |
| ETL | Extract, Load, Transform |
| FDO | Feature Data Objects |
| FIDS | Flight Information Display System |
| FPS | Fire Protection System |
| GML | Geography Markup Language |
| GIS | Geoinformation System |
| ICAO | International Civil Aviation Organization |
| ISMS | Information Security Management System |
| LOD | Level of Details |
| ODBC | Open Database Connectivity |
| OGC | Open Geospatial Consortium |
| SOA | Service-Oriented Architecture |
| SOAP | Simple Object Access Protocol |
| TETRA | Terrestrial Trunked Radio |
| TOD | Terrain and Obstacle Data |
| UAT | User Acceptance Test |
| UDDI | Universal Description, Discovery and Integration |
| WCS | Web Coverage Service |
| WFS | Web Feature Service |
| WMS | Web Map Service |

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
