# Peer review of "Location Intelligence Systems and Data Integration for Airport Capacities Planning"

_computers, doi:10.3390/computers8010013_

Round 1
Reviewer 1 Report
First, I suggest to rearrange the appearance of specific sections in the paper. This relates especially for Introduction chapter and appearance of obstacle chart on several places in the paper where it does not belong.
Further, emphasys in the paper should be on chapter 4. Application, results and discussion, and efort should be made to present developed applications and results achieved, graphically and numerically. Seams more logical for me to start this chapter with system integration, then description of applications and achieved results and at the end disucuss what could be added (BIM)...
Finally, in Conclusion statements are made which are not supported with facts in the paper. For example, in lines 458 and 459 testing and commissioning of entire system is mentioned, where in chapter 3.5 Testing, except mentioning that testing has been made no facts are given about the results of performed testing.
Therefore I suggest to revise the paper in accordance to suggestions above.
Author Response
First, I suggest to rearrange the appearance of specific sections in the paper. This relates especially for Introduction chapter and appearance of obstacle chart on several places in the paper where it does not belong.
[Authors] Done. The appearance of specific sections is rearranged. The obstacle chart with related text is moved from chapter 3. WebGIS development to chapter 1.Introduction.
Further, emphasys in the paper should be on chapter 4. Application, results and discussion, and efort should be made to present developed applications and results achieved, graphically and numerically. Seams more logical for me to start this chapter with system integration, then description of applications and achieved results and at the end disucuss what could be added (BIM)...
[Authors] The order of the sections in chapter 4 is updated in accordance with the suggestion and the chapter is expanded with additional text (Lines 434-449 and 460-469).
Finally, in Conclusion statements are made which are not supported with facts in the paper. For example, in lines 458 and 459 testing and commissioning of entire system is mentioned, where in chapter 3.5 Testing, except mentioning that testing has been made no facts are given about the results of performed testing.
[Authors] The chapter is expanded with additional text on the testing procedure (Lines 339-364).
Therefore I suggest to revise the paper in accordance to suggestions above.
Reviewer 2 Report
For the benefit of the reviewer, however, a number of points need clarifying and certain statements require further justification. My detailed recommendations are as follows:
1. Currently, such location systems and data integration systems for airport already exist on the market, and these technologies are relatively mature, so the system designed by the author lacks innovation.
2. The author takes Sarajevo Airport as an example, but the airport’s flight volume and passenger flow are smaller than some of the larger airports in the world, so the performance of the program in airports of different sizes remains to be discussed.
3. The author did not compare the program with other classic programs, so the advantages and effects of the program are not convincing. It is recommended to add some experimental simulation and data analysis.
4. In section 3.5, there are few descriptions of software testing and it is recommended to add a little to make it more convincing.
Author Response
For the benefit of the reviewer, however, a number of points need clarifying and certain statements require further justification. My detailed recommendations are as follows:
1. Currently, such location systems and data integration systems for airport already exist on the market, and these technologies are relatively mature, so the system designed by the author lacks innovation.
[Authors] The novelty is emphasized in the added text (Lines 127-152, 434-449, 555-560).
2. The author takes Sarajevo Airport as an example, but the airport’s flight volume and passenger flow are smaller than some of the larger airports in the world, so the performance of the program in airports of different sizes remains to be discussed.
[Authors] The comment was added at the end of chapter 4.1 (Lines 460-469).
3. The author did not compare the program with other classic programs, so the advantages and effects of the program are not convincing. It is recommended to add some experimental simulation and data analysis.
[Authors] Authors supplemented the paper with text in order to make its parts more convincing, but we did not make a comparison with other programs, because our goal was to demonstrate the applicability of OSS in the development of a GIS solution for small airports such as the airport Sarajevo, which had its own specific requirements.
4. In section 3.5, there are few descriptions of software testing and it is recommended to add a little to make it more convincing.
[Authors] The chapter is expanded with additional text on the testing procedure (Lines 339-364).
Reviewer 3 Report
Please see the attached file

Author Response
In this paper, the authors presented an approach of establish location intelligence system using open source software components to solve the airport infrastructure planning and construction problem. The topic is of practical interest.
However, the paper is not well written and self-contained. The existing methods in this field and their limitations were not reviewed. There are also some grammar issues in the manuscript. The authors need to address these issues before the paper can be accepted by Computers, The detailed comments are provided next.
Detailed Comments
1. The contribution or novelty should be highlighted. In the introduction section, it is not clear about what are existing methods to build the location intelligence system nor their limitations. Thus the contribution or the novelty of this paper is not clear.
[Authors] The novelty is emphasized in the added text (Lines 127-152, 434-449, 555-560).
2. There are a lot of abbreviations in this paper. Please add a nomenclature section to help the readers track those abbreviations.
[Authors] The list of abbreviation is added at the beginning of the paper.
3. In the results section, the benefit of introducing the location intelligence system should be highlighted. There lacks of metrics quantifying the advantages of using location intelligence comparing to the case of not incorporating it.
[Authors] The chapter is expanded with additional text (Lines 434-449 and 460-469).
4. Please review the grammar of the manuscript thoroughly, as there are many typo or grammar mistakes. Some examples are given below:
a. Line 9: This paper describes the an approach and methodology introduction introducing of location intelligence…
b. Line 30: delete “So”
c. Line 31: an oil distributer company
d. Line 54: Figure 2 1 /
e. Line 86. the building new terminals
f. Line 302: Thefocus of this work is the application of
g. Line 304: it is has achieved the
h. Line 439: It is the trendthat buildingloose links between software components
[Authors] The grammar of the manuscript is reviewed and the mistakes are corrected (Lines: 9, 61, 62, 85, 108, 367, 369, 388).
Round 2
Reviewer 2 Report
All the questions raised in the review report have been revised, but the English expressions need to be improved.
Reviewer 3 Report
The concerns from the reviewer have been addressed.